# Higher-Dimensional Potential Heuristics:
# Lower Bound Criterion and Connection to Correlation Complexity

**Primary Keywords:** *(4) Theory*

## Abstract

The correlation complexity introduced by Seipp et al. (2016) is a measure of a planning task indicating how hard it is. In their work, they introduced sufficient criterions to detect a correlation complexity of 2 on a planning task. They also introduced an example of a planning task with correlation complexity 3. In this work, we introduce a criterion to detect an arbitrary correlation complexity and extend the mentioned example to show with the new criterion that planning tasks with arbitrary correlation complexity exist.

## Introduction

In classical planning, we try to find a plan, a sequence of actions to transition from an initial state to a goal state. We consider satisfying planning, which looks for a plan that leads us to a goal state. In contrast, optimal planning looks for the plan with the lowest cost. Both satisfying planning and optimal planning are PSPACE-complete in general (Bylander 1994) but for some domains, it is drastically easier to find a satisfying plan than to find an optimal one.

Some satisfying planning tasks are easier to solve than others even though they have the same amount of variables and operators. This difference is hidden somewhere in the structure of the task. The correlation complexity is one measures for the difficulty of a satisfying planning task.

This measure is based on potential heuristics (Pommerening et al. 2015), which looks at the features of a given state and sums their corresponding weight to compute a heuristic value.

The correlation complexity of a task[1] $\Pi$ describes the minimal dimension of a potential heuristic $h^{pot}$ such that $h^{pot}$ is descending and dead-end avoiding. This can be interpreted as a measure of how many facts the agent has to consider at once to find the next best successor. Unless all alive states are goal states the correlation complexity is at least 1 (the agent has to consider at least *something*).

Finding the actual weights for such a potential heuristic is a hard problem and not the scope of this paper. It is PSPACE-complete in general and $\Sigma_2^p$-complete (that is the second level of the polynomial hierarchy) for heuristics with similar characteristics (Helmert et al. 2022).

---

[1] In this work we assume that each task is solvable.

In this work, we will introduce new criterions to detect a lower bound for arbitrary correlation complexity. Additionally, we prove the existence of planning tasks with arbitrary correlation complexity by applying the new criterion to gray counter tasks with an arbitrary number of bits.

## Background

We consider classical planning in the SAS$^+$ formalism. A **task** $\Pi = \langle V, I, O, \gamma \rangle$ is a tuple, where $V$ is the finite set of state variables in finite domain representation with the **domain** $dom(v)$ for each $v \in V$. We call $v \mapsto d$ with $v \in V$ and $d \in dom(v)$ a **fact**. A **partial state** $p$ is a set of facts with pairwise different variables. We denote the variables mentioned by $p$ as $vars(p) := \{v \mid v \mapsto d \in p\}$. A **state** $s$ is a partial state containing all state variables (i.e. $vars(s) = V$), we use the notation $s \in \Pi$ to indicate that $s$ is a state of the task $\Pi$. $I$ is the **initial state**, and $\gamma$ is a partial state indicating the **goal** of the task.

The **projection** of a partial state $s$ to the state variables $W \subseteq vars(s)$ is denoted as $s^W := \{v \mapsto d \in s \mid v \in W\}$.

Each **operator** $o = \langle pre(o), eff(o) \rangle$ in $O$ contains a **precondition** and an **effect**. Both of them are partial states. An operator $o$ is **applicable** in state $s$ if $pre(o) \subseteq s$. The **successor** of a state $s$ and an applicable operator $o$ is the state $s[\![o]\!]$ with $s[\![o]\!]^{vars(eff(o))} = eff(o)$ and $s[\![o]\!]^{V \setminus vars(eff(o))} = s^{V \setminus vars(eff(o))}$. We use the notation $s[\![o!]\!]$ to **forcefully** apply operator $o$ to state $s$, with $s[\![o!]\!] := s[\![\langle \emptyset, eff(o) \rangle]\!]$. We assume tasks to be in **normal form**, meaning $vars(eff(o)) \subseteq vars(pre(o))$ and $eff(o) \cap pre(o) = \emptyset$ for each operator $o \in O$.

We say two operators $o, o'$ are **inverse** of each other if $vars(eff(o)) = vars(eff(o'))$, $eff(o) \subseteq pre(o')$ and $eff(o') \subseteq pre(o)$. Intuitively, everything changed by applying $o$ to a state where it is applicable changes back by (forcefully) applying $o'$ to that successor.

A list of operators $m = [o_1, \ldots, o_n]$ is called a **macro**, we treat them the same as operators with $pre([o_1, o_2]) = (pre(o_2)eff(o_1)) \cup pre(o_1)$ and $eff([o_1, o_2]) = eff(o_2) \cup \{v \mapsto d \in eff(o_1) \mid v \notin vars(eff(o_2))\} \setminus pre([o_1, o_2])$. Additionally, $[o_1, o_2, o_3, \ldots, o_n] = [[o_1, o_2], o_3 \ldots, o_n]$ is a macro, too.

We can **recompose** a macro by representing it as a different but equivalent list of macros and/or operators that keeps the same underlying list of operators.

For example $[[o_1, o_2, o_3], o_4, [o_5]]$ can be recomposed into $[[o_1, o_2], [o_3, o_4], [o_5]]$. We call the macros and operators in the outermost list the **base macros**.

Applying a singleton macro $[o]$ is the same as applying the underlying operator. Applying a macro $[m_1, m_2, \ldots, m_n]$ to state $s$ is the same as applying the base macros $m_1, m_2, \ldots, m_n$ in succession to $s$, meaning $s[[m_1, m_2, \ldots, m_n]] = s[[m_1]][[m_2, \ldots, m_n]]$.

The set $pre([o_1, o_2])$ might be no partial state. Unless explicitly mentioned we only talk about non-empty macros.

If a macro $m$ is applicable in $s$ and $\gamma \subseteq s[[m]]$, then $m$ is called an $s$-**plan** (such marcos might be empty).

We call a state $s$ **solvable** if an $s$-plan exists. A state is called **alive** if a macro $m$ exists that is applicable in $I$, $I[[m]] = s$ and $s$ is solvable.

A **heuristic** $h(s)$ maps each state $s$ to a value in $\mathbb{R} \cup \{\infty\}$, indicating how "good" the state is for the search (the lower the better).

A heuristic $h$ is **descending** if each non-goal alive state $s$ has a successor $s'$ with $h(s) > h(s')$. A heuristic $h$ is **dead-end avoiding** if it holds for each non-goal alive state $s$ that each successor $s'$ of $s$ with $h(s) > h(s')$ is solvable (Seipp et al. 2016).

With $[n, m]$ we denote the set $\{x \mid x \in \mathbb{N}, n \leq x \leq m\}$.

### Potential Heuristic

Potential heuristics are a class of heuristics introduced by Pommerening et al. (2015). A **potential heuristic** is a function $h_{pot} : \mathcal{P} \to \mathbb{R} \cup \{\infty\}$ that is computed with a weighted count of the partial assignments that agree with the given partial assignments.

$$h_{pot}(s) = \sum_{p \in \mathcal{P}} (w(p) \cdot [p \subseteq s])$$

where $\mathcal{P}$ is the set of all possible partial assignments for the task, $[p \subseteq s]$ is in the Iverson bracket notation, and $w(p)$ is the **weight** for the partial assignment $p$. In practice, most of the weights are 0. The **dimension** of a potential heuristic is $max_{p \in P, w(p) \neq 0} |p|$.

### Correlation Complexity

Correlation complexity is a measure of the complexity of planning tasks introduced by Seipp et al. (2016) which is based on potential heuristics. They looked for potential heuristics that are descending and dead-end avoiding (DDA).

**Definition 1** (Correlation Complexity). *The **correlation complexity** of a planning task $\Pi$ is defined as the minimal dimension of all descending, dead-end avoiding potential heuristics for $\Pi$. The correlation complexity of a domain is the maximal correlation complexity over all tasks in that domain.*

Finding a plan with the use of a DDA heuristic is easily done with simple hill-climbing. This algorithm works by starting at the initial state and repeatedly updating the current state with a successor that has a lower heuristic value until a goal state is found. This can be interpreted as wandering downwards in the state space. The number of state expansions is bounded by $h^{pot}(I) - \min_{s \in S} h^{pot}(s)$ if all weights are integers (Seipp et al. 2016).

Seipp et al. investigated multiple IPC planning domains and showed that all of them had a correlation complexity of 2, but it is possible to construct planning tasks with correlation complexity of 3.

## Lower Bounds of the Correlation Complexity

We want to understand better what causes a large correlation complexity. Seipp et al. (2016) introduced two criterions to identify that a task has correlation complexity of at least 2. They are based on critical and dangerous operators.

**Definition 2.** *Let $\Pi = \langle V, I, O, \gamma \rangle$. An operator $o$ is **critical** in task $\Pi$ if there is an alive state $s \in \Pi$ such that $\langle V, s, O \setminus \{o\}, \gamma \rangle$ is unsolvable. An operator $o$ is **dangerous** in task $\Pi = \langle V, I, O, \gamma \rangle$ if there is an alive state $s \in \Pi$ with $o$ applicable in $s$, such that $\langle V, s[[o]], O, \gamma \rangle$ is unsolvable.*

**Theorem 1.** *Let $\Pi$ be a planning task in normal form, and let $o$ and $o'$ be critical operators of $\Pi$ that are inverses of each other. Then $\Pi$ has correlation complexity of at least $2$*

**Theorem 2.** *Let $\Pi$ be a planning task in normal form, and let $o$ be an operator that is critical and dangerous in $\Pi$. Then $\Pi$ has correlation complexity of at least $2$*

These criterions focus on the operators in the task. We want to add criterions that look from a different angle and focus on the states instead.

## Quadruple Criterion

We introduce the quadruple criterion. Roughly speaking it checks if a given potential heuristic can be represented as a 2-dimensional heuristic. Note that any heuristic can be translated into a potential heuristic if the dimension is large enough.

**Theorem 3** (Quadruple Criterion). *Let $\Pi = \langle V, I, O, \gamma \rangle$ be a planning task, and let $h^{pot}$ be a potential heuristic. If there exist states $a, b, c, d$ in $\Pi$ and a partition $\{W, M\}$ of $V$ such that:*

$$h^{pot}(a) > h^{pot}(b), \quad a^W = b^W, \quad a^M = d^M,$$
$$h^{pot}(c) \geq h^{pot}(d), \quad c^W = d^W, \quad b^M = c^M,$$

*then the dimension of $h^{pot}$ is at least 2.*

For the proof we use the fact that we can split one (finite) sum over all $x \in X$ with a partition $\{Y, Z\}$ of $X$ into two sums where one iterates over $y \in Y$ and the other over $z \in Z$.

$$\sum_{x \in X} f(x) = \sum_{y \in Y} f(y) + \sum_{z \in Z} f(z),$$

with $f$ an arbitrary function.

*Proof.* Let $h^{pot}$ be a heuristic and $a, b, c, d$ states in $\Pi$ and $\{W, M\}$ a partition of $V$ such that:
$h^{pot}(a) > h^{pot}(b)$ and $h^{pot}(c) \geq h^{pot}(d)$ and $a^W = b^W$ and $c^W = d^W$ and $a^M = d^M$ and $b^M = c^M$.

Assume that the dimension of the potential heuristic $h^{pot}$ is at most 1. The assumption implies that $h^{pot}(s) = \sum_{p \in \mathcal{P}, |p| \leq 1} w(p) \cdot [p \subseteq s] = \sum_{p \in \mathcal{P}, |p| = 1, vars(p) \subseteq W} w(p) \cdot$

$[p \subseteq s^W] + \sum_{p \in \mathcal{P}, |p|=1, vars(p) \subseteq M} w(p) \cdot [p \subseteq s^M] + w(\emptyset) = h^{pot}(s^W) + h^{pot}(s^M) - w(\emptyset)$ for each state $s$. The weight $w(\emptyset)$ is subtracted because $\emptyset \subseteq s^W$ and $\emptyset \subseteq s^M$. This provides us with the two inequalities

$$h^{pot}(a^W) + h^{pot}(a^M) - w(\emptyset) > h^{pot}(b^W) + h^{pot}(b^M) - w(\emptyset)$$

$$h^{pot}(c^W) + h^{pot}(c^M) - w(\emptyset) \geq h^{pot}(d^W) + h^{pot}(d^M) - w(\emptyset).$$

Since $a^W = b^W$ and $c^W = d^W$ we can simplify the inequalities to

$$h^{pot}(a^M) > h^{pot}(b^M)$$

$$h^{pot}(c^M) \geq h^{pot}(d^M).$$

We know that $a^M = d^M$ and $b^M = c^M$ so we replace these values in the latest inequality and end up with

$$h^{pot}(a^M) > h^{pot}(b^M)$$

$$h^{pot}(b^M) \geq h^{pot}(a^M).$$

which is a contradiction. So the assumption is not true and therefore the dimension of the potential heuristic $h^{pot}$ is at least 2. □

The conditions of the quadruple criterion are sufficient to detect a dimension of at least 2. Is it also a necessary condition? The answer is no.

Consider the two heuristics on a task with two variables of binary domain (with $'XY'$ we denote the state $\{v_1 \mapsto X, v_2 \mapsto Y\}$): $h_1('00') = 0$, $h_1('10') = 40$, $h_1('01') = 2$, $h_1('11') = 42$ and $h_2$ which is equal to $h_1$ except for $h_2('11') > 42$. It is easy to see that $h_1$ is of dimension 1 with $w_1(v_1) = 40$, $w_1(v_2) = 2$ but $h_2$ is not. However, the order relations $(>, \geq)$ are the same. Meaning, that if we could find an assignment of $a, b, c, d, W, M$ for $h_2$ to detect its larger dimension by Theorem 4 we could use the same for $h_1$, but that is not possible as $h_1$ is of dimension 1.

We can use the quadruple criterion to argue about the correlation complexity.

**Theorem 4.** *Let $\Pi = \langle V, I, O, \gamma \rangle$ be a planning task. If for each potential heuristic $h^{pot}$ that is DDA on $\Pi$ there exist states $a, b, c, d$ in $\Pi$ and a partition $\{W, M\}$ of $V$ such that:*

$$h^{pot}(a) > h^{pot}(b), \quad a^W = b^W, \quad a^M = d^M,$$

$$h^{pot}(c) \geq h^{pot}(d), \quad c^W = d^W, \quad b^M = c^M,$$

*then the correlation complexity of $\Pi$ is at least 2.*

Note that these states and the partition do not have to be the same for all DDA potential heuristics.

*Proof.* We know that the correlation complexity of a task $\Pi$ is the minimal dimension over all potential heuristics that are DDA on $\Pi$. The condition of Theorem 4 implies that the quadruple criterion is applicable on each DDA potential heuristic on $\Pi$. Therefore, we can apply the quadruple criterion on each potential heuristic that is DDA on $\Pi$. We conclude that the dimension of each potential heuristic that is DDA on $\Pi$ is at least 2. Therefore, the correlation complexity of $\Pi$ is at least 2. □

Theorem 4 is a generalization of the criterion from Theorem 2 that Seipp et al. (2016) used to detect correlation complexity of at least 2 (proof in Appendix A). Theorem 3 is equivalent to the first (non-trivial) instance of a pattern, which we will call the $2^n$ *states criterion*. Here we formulate and prove it. Later, we will use the $2^n$ states criterion as a stepping stone to construct the *macro folding criterion*, a generalization of the criterion from Theorem 1. We then apply it to the family of *Gray counter tasks* and show that a Gray counter task has a correlation complexity equal to its number of bits.

## $2^n$ states criterion

We first state some definitions, notations, and lemma. Thereby we will introduce the $\Pi^{\leq k}$ construction to formulate the $2^n$ states criterion. The $\Pi^{\leq k}$ construction is similar to the $P^m$ construction (Haslum 2009), to the $\Pi^{\mathcal{C}}$ compilation (Steinmetz and Hoffmann 2018), and to fluent merging (van den Briel, Kambhampati, and Vossen 2007). We discuss the similarities and differences in the related work section.

The $\Pi^{\leq k}$ construction assumes an arbitrary order of the state variables.

**Definition 3** ($\Pi^{\leq k}$ construction). *Let $\Pi = \langle V, I, O, \gamma \rangle$ a planning task. Let $V$ be arbitrarily ordered, $V = \{v_1, \ldots, v_{|V|}\}$. Let $s$ be a state and $k \leq |V|$.*

*Let $p \supseteq \{\langle v_{i_1}, d_{i_1} \rangle, \ldots, \langle v_{i_k}, d_{i_k} \rangle\}$ be a partial state with $i_a < i_b$ for each $0 < a < b \leq k$.*

*We call $meta_{\{i_1, \ldots, i_k\}}(p) = \langle v_{\langle i_1, \ldots, i_k \rangle}, \langle d_{i_1}, \ldots, d_{i_k} \rangle \rangle$ a* **metafact** *with the* **metavariable** *$v_{\langle i_1, \ldots, i_k \rangle}$. The domain of the metavariable is $\times_{j=1}^k dom(v_{i_j})$. The* **size** *of the metafact/metavariable is $k$.*

*We say metavariable $v_l$ is* **created** *from variable $v_t$ if $t \in l$, we refer to these variables with $creators(v_l)$.*

*With $meta^{\leq k}(s) = \bigcup_{S \subseteq V, |S| \leq k} \{meta_S(s)\}$ we denote the set of all metafacts from $s$ of size up to $k$. We call $meta^{\leq k}(s)$ the* **metastate** *of size $k$ of $s$ and use the shorthand notation $s^{\leq k}$ and the set of its metavariables as $V^{\leq k}$. The $\Pi^{\leq k}$* **construction** *is the set $\{meta^{\leq k}(s) | s \in \Pi\}$.*

Let us look at an example. In the Termes domain a robot has to build towers out of blocks on a 2D grid map. These blocks are also used to build stairs to reach the top of the towers. The robot can move to neighboring cells that contain a tower at most one level apart from the level the robot is at. It can change the height of a neighboring tower by placing/removing a block. A block can only be placed if the robot carries one and the neighboring tower is on the same level as the robot. Removing a block is only possible if the neighboring tower is one level above the robot and it is not carrying anything. Additionally, the robot can create/destroy blocks if it is at the deposit cell. It is not allowed to place blocks on this cell.

Consider a task with 4 cells, one of them the deposit, and a maximal tower height of 3. The state variables (without the constants, ordered as written) is

$$V = \{v_{height2}, v_{height3}, v_{height4}, v_{robotAt}, v_{robotHand}\},$$

with domain $\{0, 1, 2, 3\}$ for $v_{height2}, v_{height3}$ and $v_{height4}$, $dom(v_{robotAt}) = \{cell1, cell2, cell3, cell4\},$

and $dom(v_robotHand) = \{free, full\}$. Let initial state be

$$I = \{v_{height2} \mapsto 1, v_{height3} \mapsto 1, v_{height4} \mapsto 3,$$
$$v_{robotAt} \mapsto cell2, v_{robotHand} \mapsto free\}.$$

The corresponding metastate $I^{V^{\leq 2}}$ contains 31 metafacts and is a superset of

$$\{v_{\langle\rangle} \mapsto \langle\rangle, v_{\langle height2\rangle} \mapsto \langle 1\rangle, v_{\langle robotHand\rangle} \mapsto \langle free\rangle,$$
$$v_{\langle robotAt, robotHand\rangle} \mapsto \langle cell2, free\rangle,$$
$$v_{\langle height2, height4\rangle} \mapsto \langle 1, 3\rangle\}.$$

We have facts in the metastate that encode multiple facts at once. With the metastates, we can artificially reduce the dimension of a potential heuristic.

**Definition 4** (*k-synchronized heuristic*). *Let $k \in \mathbb{N}_0$, $\Pi$ a planning task. A function $h' : \Pi^{\leq k} \to \mathbb{R}$ is a k-**synchronized heuristic** of the heuristic $h$ on $\Pi$ if $h'(s^{\leq k}) = h(s)$ for each state $s$ in $\Pi$.*

**Lemma 1.** *Let $\Pi = \langle V, I, O, \gamma\rangle$ be a planning task. If there exists a potential heuristic $h^{pot}$ of dimension $k$ on $\Pi$ then there exists a k-synchronized potential heuristic $h'^{pot}$ of dimension 1.*

*Proof.* We remember that $h^{pot}(s) = \sum_{p \in \mathcal{P}}(w(p) \cdot [p \subseteq s])$. For each partial assignment $p = \{v_{i_1} \mapsto d_{i_1}, \dots, v_{i_m} \mapsto d_{i_m}\}$ with $|p| \leq k$ and $p \subseteq s$ there exists a corresponding metafact $f_p = (v_{\langle i_1, \dots, i_m\rangle} \mapsto \langle d_{i_1}, \dots, d_{i_m}\rangle)$ in the metastate $s^{\leq k}$ and therefore a partial assignment $p^* = \{f_p\}$ of size 1. Let $\mathcal{P}^*$ be the set of all possible partial assignments of size 1 in $\Pi^{\leq k}$. By choosing $w'(p^*) = w(p)$ for each $p$ we see that $h^{pot}(s) = \sum_{p \in \mathcal{P}}(w(p) \cdot [p \subseteq s]) = \sum_{p^* \in \mathcal{P}^*}(w'(p^*) \cdot [p^* \subseteq s^{\leq k}]) = h'^{pot}(s^{\leq k})$.

$\square$

Looking at the contrapositive of Lemma 1 we see that we can use the metastates of size $k$ to check if it is impossible to represent a given heuristic as a potential heuristic of dimension $k$. We will use this to prove the $2^n$ states criterion.

In the following, we always use the metastates to be projected. If a subset $D \subseteq V^{\leq k}$ is used for the mapping of a state $s$ to a different set of metavariables $D$ we mean with $s^D$ the metastate $s^{\leq k}$ projected to $D$, denoted as $s^D := (s^{\leq k})^D$.

**Theorem 5** ($2^n$ *States Criterion*). *Let $n \in \mathbb{N}_1$, $\Pi = \langle V, I, O, \gamma\rangle$ be a planning task, $h$ be a potential heuristic on $\Pi$. If there exist states $s_0, \dots, s_{2^n-1}$ in $\Pi$ and a partition $\{D_0, \dots, D_{n-1}\}$ of $V^{\leq n-1}$ and $j^* \in [0, 2^{n-1} - 1]$ such that:*

$$h(s_{2 \cdot j^*}) > h(s_{2 \cdot j^*+1})$$

$$h(s_{2 \cdot j}) \geq h(s_{2 \cdot j+1}) \text{ for all } j \in [0, 2^{n-1} - 1] \setminus \{j^*\}$$

*and*

$$\forall d \in [0, n-1] \forall g \in [0, 2^{n-1-d} - 1] \forall i \in [0, 2^d - 1] :$$
$$s_{g \cdot 2^{d+1}+i}^{D_d} = s_{(g+1) \cdot 2^{d+1}-1-i}^{D_d} \quad (1)$$

*then the dimension of $h$ is at least $n$.*

| | | | | | | | | | |
|---|---|---|---|---|---|---|---|---|---|
| $d = 3$ | $g$ | 0 | | | | | | | |
| | $i$ | 0 | 1 | 2 | 3 | 4 | 5 | 6 | 7 |
| | $s_{LHS}$ | 0 | 1 | 2 | 3 | 4 | 5 | 6 | 7 |
| | $s_{RHS}$ | 15 | 14 | 13 | 12 | 11 | 10 | 9 | 8 |
| $d = 2$ | $g$ | 0 | | | | 1 | | | |
| | $i$ | 0 | 1 | 2 | 3 | 0 | 1 | 2 | 3 |
| | $s_{LHS}$ | 0 | 1 | 2 | 3 | 8 | 9 | 10 | 11 |
| | $s_{RHS}$ | 7 | 6 | 5 | 4 | 15 | 14 | 13 | 12 |
| $d = 1$ | $g$ | 0 | | 1 | | 2 | | 3 | |
| | $i$ | 0 | 1 | 0 | 1 | 0 | 1 | 0 | 1 |
| | $s_{LHS}$ | 0 | 1 | 4 | 5 | 8 | 9 | 12 | 13 |
| | $s_{RHS}$ | 3 | 2 | 7 | 6 | 11 | 10 | 15 | 14 |
| $d = 0$ | $g$ | 0 | 1 | 2 | 3 | 0 | 1 | 2 | 3 |
| | $i$ | 0 | 0 | 0 | 0 | 0 | 0 | 0 | 0 |
| | $s_{LHS}$ | 0 | 2 | 4 | 6 | 8 | 10 | 12 | 14 |
| | $s_{RHS}$ | 1 | 3 | 5 | 7 | 9 | 11 | 13 | 15 |

Table 1: Example values for $n = 4$ of condition (1).

Before we prove the $2^n$ states criterion we want to shed light on the interpretation of condition (1). For a fixed $d = n - 1$ one can consider the states $s_0, \dots, s_{2^n-1}$ split into 2 parts. The first part contains $s_0, \dots, s_{2^{n-1}-1}$ and the second part contains $s_{2^{n-1}}, \dots, s_{2^n-1}$. Variable $g$ stays 0, only $i$ iterates. The left-hand side of the equation iterates through the first part forwards and the right-hand side through the second part backward.

For $d = n - 2$, one can consider the two parts split again in the middle resulting in 4 parts. With $g = 0$, the left-hand side of the equation iterates through the first part forward and the second part backward. For $g = 1$, the left-hand side of the equation iterates through the third part forward and the fourth part backward.

For decreasing $d$ the number of such parts doubles and the left-hand side iterates through the odd parts forward and the right-hand side iterates through the even parts backwards. Until $d = 0$, there we produce half as many parts as states. Variable $i$ does not iterate, only $g$ does. The left-hand side of the equation iterates through the states with even index and the right-hand side through the states with odd index.

Table 1 shows the example values for $n = 4$. Looking at the red indicated numbers we see that state $s_6$ has to be equal to: (i) $s_9$ under projection to $D_3$, (ii) $s_1$ under projection to $D_2$, (iii) $s_5$ under projection to $D_1$, and (iv) $s_7$ under projection to $D_0$.

*Proof.* Assume there exists a potential heuristic $h'$ of dimension 1 on the task $\Pi^{\leq n-1}$, with $h'$ being an $n - 1$-synchronized heuristic to $h$, and a partition $\{D_0, \dots, D_{n-1}\}$ of $V^{\leq n-1}$ and $j^* \in [0, 2^{n-1} - 1]$ such that:

$$h(s_{2 \cdot j^*}) > h(s_{2 \cdot j^*+1}),$$
$$h(s_{2 \cdot j}) \geq h(s_{2 \cdot j+1}) \text{ for all } j \in [0, 2^{n-1} - 1] \setminus \{j^*\} \quad (2)$$

and

$$\forall d \in [0, n-1] \forall g \in [0, 2^{n-1-d} - 1] \forall i \in [0, 2^d - 1] :$$
$$s_{g \cdot 2^{d+1}+i}^{D_d} = s_{(g+1) \cdot 2^{d+1}-1-i}^{D_d}$$

We consider the sum of all the inequalities from (2):

$$\sum_{j=0}^{2^{n-1}-1} h(s_{2 \cdot j}) > \sum_{j=0}^{2^{n-1}-1} h(s_{2 \cdot j+1})$$

We see that the left-hand side only contains states with an even index and the right-hand side contains only states with an odd index. With $h'$ being of dimension 1 and $n-1$-synchronized to $h$ we can split $h(s_t) = \sum_{d=0}^{n-1} h'(s_t^{D_d}) - \sum_{d=0}^{n-2} w(\emptyset)$ and obtain

$$\sum_{j=0}^{2^{n-1}-1} \sum_{d=0}^{n-1} h'(s_{2 \cdot j}^{D_d}) > \sum_{j=0}^{2^{n-1}-1} \sum_{d=0}^{n-1} h'(s_{2 \cdot j+1}^{D_d}) \quad (3)$$

Consider $s_{g \cdot 2^{d+1}+i}^{D_d} = s_{(g+1) \cdot 2^{d+1}-1-i}^{D_d}$ for an arbitrary $d$. The index on the left-hand side is even iff $i$ is even. The index on the right-hand side is odd iff $i$ is even.

We conclude that for every fixed $d$ and for every odd $x$ with $x \in [0, 2^n-1]$ there exists one even $y$ with $y \in [0, 2^n-1]$ such that

$$s_x^{D_d} = s_y^{D_d} \quad (4)$$

With equation (4) we can iterate through all $d$ with $d \in [0, n-1]$ and all even numbers $p$ with $p \in [0, 2^n-1]$. We find a corresponding odd $q$ with $q \in [0, 2^n-1]$ such that $s_p^{D_d} = s_q^{D_d}$ and reduce (3) by removing $h'(s_p^{D_d})$ on the left-hand side and $h'(s_q^{D_d})$ on the right-hand side. This change does not affect the inequality. However, all summands are removed and we get the inequality $0 > 0$ which is a contradiction.

We conclude that there exists no $n-1$-synchronized potential heuristic to $h$ of dimension 1. With the contrapositive of Lemma 1, we conclude that no potential heuristic $h$ on $\Pi$ with dimension $n-1$ exists. Therefore, the dimension of $h$ is at least $n$. $\qquad \square$

The $2^2$ states criterion is equivalent to the quadruple criterion with the minor detail that $\{D_0, D_1\}$ is a partition of $V^{\leq 1}$, unlike $\{W, M\}$, which is a partition of $V$. The difference is minor as $V^{\leq 1}$ only contains meta variables of size 1 and $v_{\langle\rangle}$, which is constant.

The $2^1$ states criterion projects the 2 states to the set $\{v_{\langle\rangle}\}$. There, all states are the same, trivially. What remains is the condition that there are 2 states with different heuristic values.

**Theorem 6.** *Let $n \in \mathbb{N}_1$, $\Pi = \langle V, I, O, \gamma \rangle$ be a planning task. If for each potential heuristic $h^{pot}$ that is DDA on $\Pi$ there exist states $s_0, \ldots, s_{2^n-1}$ in $\Pi$, a partition $\{D_0, \ldots, D_{n-1}\}$ of $V^{\leq n-1}$ and $j^* \in [0, 2^{n-1}-1]$ such that:*

$$h(s_{2 \cdot j^*}) > h(s_{2 \cdot j^*+1}),$$

$$h(s_{2 \cdot j}) \geq h(s_{2 \cdot j+1}) \text{ for all } j \in [0, 2^{n-1}-1] \setminus \{j^*\},$$

*and*

$$\forall d \in [0, n-1] \forall g \in [0, 2^{n-1-d}-1] \forall i \in [0, 2^d-1]:$$

$$s_{g \cdot 2^{d+1}+i}^{D_d} = s_{(g+1) \cdot 2^{d+1}-1-i}^{D_d}$$

*then the correlation complexity of $\Pi$ is at least $n$.*

Note that these states and the partition do not have to be the same for all DDA potential heuristics.

*Proof.* We know that the correlation complexity of a task $\Pi$ is the minimal dimension over all potential heuristics that are DDA on $\Pi$. The condition of Theorem 6 implies that any DDA potential heuristic fulfills the condition for the $2^n$ states criterion. Therefore, we conclude that each potential heuristic that is DDA on $\Pi$ is of dimension at least $n$. Therefore, the correlation complexity of $\Pi$ is at least $n$. $\qquad \square$

## Folded Macro Criterion

With the shifted view from operators to states we were able to construct a family of criterions to detect arbitrary correlation complexity. Now we want to shift the view back to operators and macros as they are in some sense more accessible for higher-level arguments. We create a generalization of the criterion from Theorem 1.

**Definition 5** (Macro Folding). *The macros $\overrightarrow{m}, \overleftarrow{m}$ are $n$-matching if we can decompose both into $n$ base macros, where the $i$-th base macro of $\overrightarrow{m}$ is the inverse of the $n-i+1$-th base macro of $\overleftarrow{m}$ for each $i \in [1, n]$.*

*We say a macro $m$ is **folded** one time on **crease** $\widehat{m}_1$ if we can decompose it into 3 macros $[\overrightarrow{m}_0, \widehat{m}_1, \overleftarrow{m}_0]$ with $\overrightarrow{m}_0$ 1-matching $\overleftarrow{m}_0$. Here we call $\widehat{m}_1$ the first crease and $\overrightarrow{m}_0$ the 0-th crease. We say a macro $m$ is folded $n > 1$ times if it is folded once on crease $\widehat{m}_n$ (the $n$-th crease of $m$) and we can decompose it into 7 macros $[\overrightarrow{m}_{n-1}, \widehat{m}_{n-1}, \overleftarrow{m}_{n-1}, \widehat{m}_n, \overrightarrow{m}'_{n-1}, \widehat{m}_{n-1}, \overleftarrow{m}'_{n-1}]$ where $[\overrightarrow{m}_{n-1}, \widehat{m}_{n-1}, \overleftarrow{m}_{n-1}]$ and $[\overrightarrow{m}'_{n-1}, \widehat{m}_{n-1}, \overleftarrow{m}'_{n-1}]$ are $2^n-1$-matching, folded $n-1$ times on crease $\widehat{m}_{n-1}$ and $\widehat{m}_{n-1}$ respectively. Both, $\widehat{m}_{n-1}$ and $\widehat{m}_{n-1}$ are the $n-1$-th creases of $m$, with $creases(m, n-1)$ we refer to the set of $n-1$-th creases of $m$.*

Note that if a macro $m$ is folded $n$ times it is also folded $k$ times for all $0 \leq k \leq n$. Asking for the $k$-th creases is therefore ill-defined, without specifying how many total creases are considered. In the following, we implicitly mean the largest possible value we established for the considered macro.

An intuitive interpretation of a folded macro $[\overrightarrow{m}, \widehat{m}, \overleftarrow{m}]$ is to view $\overrightarrow{m}$ as the set-up and $\overleftarrow{m}$ as the tear-down. The crease $\widehat{m}$ is the actually desired action. If the macro is folded multiple times there are recursive occurrences of this set-up and tear-down behavior.

The Termes task described earlier with goal $\{v_{height2} \mapsto 1, v_{height3} \mapsto 1, v_{height4} \mapsto 2, \}$ is an example for that. The task is to remove the 3rd block from cell 4 (and keep the other towers the same). Note that the encoding does not distinguish the individual blocks.

The set-up for that is to carry a block to cell 2. From there the robot can place that block on cell 3, climb on it, pick up the block from cell 4, and carry it to cell 2. We view this macro of 4 operators as the crease. Afterward, the tear-down is to get rid of the auxiliary block.

To execute the set-up a secondary set-up is needed. The robot has to move to the deposit first (secondary set-up), create a block (crease of the primary set-up), and move back up with it (secondary tear-down).

Similarly, for the tear-down. A secondary set-up is needed there, as well. The robot has to carry the block it holds to the deposit (secondary set-up), destroy the block (crease of the primary tear-down), and move back up (secondary tear-down).

The robot is required to come back to cell 2 after destroying the carried block, to pick up the auxiliary block. The described order of operators is the only one solving the task (without cycles or transitions after reaching the goal). Therefore, the macro of the primary set-up and primary tear-down are critical.

We provided this task in PDDL in Appendix B together with the plan and a visualization of the folded macro.

**Lemma 2** (Orthogonality Lemma). *For any state $s$ and applicable, 1 time folded macro $m = [\overrightarrow{m}_n, \ldots, \overrightarrow{m}_0, \widehat{m}, \overleftarrow{m}_0, \ldots, \overleftarrow{m}_n]$ with crease $\widehat{m}$ it holds for all $i \in [0, n]$:*

- $s[\![\overrightarrow{m}_i!]\!][\![\widehat{m}!]\!] = s[\![\widehat{m}!]\!][\![\overrightarrow{m}_i!]\!]$,
- $vars(\textit{eff}(\overrightarrow{m}_i)) \cap vars(\textit{eff}(\widehat{m})) = \emptyset$, *and*
- $s[\![m]\!] = s[\![\widehat{m}!]\!]$.

*Proof.* Case (i) $i = 0$: Assume $s[\![\overrightarrow{m}_i!]\!][\![\widehat{m}!]\!] \neq s[\![\widehat{m}!]\!][\![\overrightarrow{m}_i!]\!]$, therefore there is a $v \in vars(\textit{eff}(\overrightarrow{m}_i)) \cap vars(\textit{eff}(\widehat{m}))$, and $v \mapsto d_A$ is an effect of $\overrightarrow{m}_i$ and $v \mapsto d_B$ is an effect of $\widehat{m}$. We know $d_A \neq d_B$, due to our assumption. Then $\overleftarrow{m}_i$ is not applicable in state $s[\![\overrightarrow{m}_i]\!][\![\widehat{m}]\!]$ as it has $v \mapsto d_A$ as precondition, because it is in normal form and inverse to $\overrightarrow{m}_i$. This provides a contradiction to $m$ being an applicable macro. This also provides us that $s[\![\overrightarrow{m}_0!]\!][\![\widehat{m}!]\!][\![\overleftarrow{m}_0!]\!] = s[\![\widehat{m}!]\!]$.

Case (ii) $i > 0$: Consider the macro $[\overrightarrow{m}_{i-1}, \ldots, \overrightarrow{m}_0, \widehat{m}, \overleftarrow{m}_0, \ldots, \overleftarrow{m}_{i-1}]$ as the crease of $m$ and relabel the indices to match case (i). $\qquad\square$

A variable $v$ might appear in the effects of $o \in \widehat{m}$ and $o' \in \overrightarrow{m}_i$. The effect just has to be undone again in one of the base macros $\widehat{m}$ or $\overrightarrow{m}_i$, resulting in a 'defacto' effect of the macro where $v$ does not appear.

**Lemma 3** (Matching Lemma). *Let $n \in \mathbb{N}$, $s_0$ be a state where the macro $m := [\overrightarrow{m}, \widehat{m}, \overleftarrow{m}]$ is applicable and $\overrightarrow{m}, \overleftarrow{m}$ are $n - 1$-matching. Let $s_i := s_{i-1}[\![\overrightarrow{m}_i]\!]$ with $\overrightarrow{m}_i$ being the $i$-th base macro of $\overrightarrow{m}$ for $i \in [1, n-1]$, $s_n := s_{n-1}[\![\widehat{m}]\!]$, and $s_{n+i} := s_{n+i-1}[\![\overleftarrow{m}_i]\!]$ with $\overleftarrow{m}_i$ being the $i$-th base macro of $\overleftarrow{m}$ for $i \in [1, n-1]$. Then for all $j \in [0, n-1]$*

$$s_{n-1-j}[\![\widehat{m}!]\!] = s_{n+j}.$$

*Proof.* Let $j \in [0, n-1]$. Since $m$ is folded on crease $\widehat{m}$ we conclude with Lemma 2 that $s_{n-1-j}[\![\widehat{m}!]\!] = s_{n-1-j}[\![[\overrightarrow{m}_{n-j+1}, \ldots, \overrightarrow{m}_n, \widehat{m}, \overleftarrow{m}_n \ldots \overleftarrow{m}_{n-j+1}]\!]\!] = s_{n+j}$ by construction. $\qquad\square$

With that, we introduce the new criterion.

**Theorem 7** (Folded Macro Criterion). *Let $\Pi$ be a planning task in normal form, and let $\overrightarrow{m}$ and $\overleftarrow{m}$ be critical macros of $\Pi$ that are $2^n - 1$-matching and folded $n - 1$ times, then $\Pi$ has correlation complexity of at least $n + 1$.*

*Proof.* Since $\overrightarrow{m}$ and $\overleftarrow{m}$ are critical each plan for $\Pi$ is of the form $[m^I, \overrightarrow{m}, \widehat{m}, \overleftarrow{m}, m^\gamma]$ (or $\overrightarrow{m}, \overleftarrow{m}$ swapped but without loss of generality we assume the former; $m^I$ and/or $m^\gamma$ might be empty) and therefore contains the macro $m := [\overrightarrow{m}, \widehat{m}, \overleftarrow{m}]$. This macro is folded $n$ times, with the $n$-th crease at $\widehat{m}$.

Let us consider the $2^{n+1}$ states $s_0, \ldots, s_{2^{n+1}-1}$ we can extract from this $n$ times folded macro. With $s_0 = I[\![m^I]\!]$ and $s_i = s_{i-1}[\![b_i]\!]$, where $b_i$ is the $i$-th base macro of $m$.

Obviously, for each DDA heuristic $h$ on $\Pi$ there exists a choice of $m^I, \widehat{m}, m^\gamma$ such that it holds:

$$h(s_{2 \cdot j}) > h(s_{2 \cdot j+1}) \text{ for all } j \in [0, 2^n - 1]$$

Matching the first condition of the $2^{n+1}$ state criterion.

For the second condition we have to find a fitting partition of $V^{\leq n}$.

We consider $D^{m:n} = \{D_0^{m:n}, \ldots, D_n^{m:n}\}$ with $D_{n-i}^{m:n} := \{v \in V^{\leq n} \setminus \bigcup_{j=n-i+1}^n D_j^{m:n} \mid creators(v) \cap vars(\textit{eff}(\widehat{m}_{n-i})) = \emptyset\}$ with $\widehat{m}_{n-i} \in creases(m, n - i)$. These sets are obviously not intersecting. None of them is empty because the meta-variable $v^{n-i}$ with $creators(v^{n-i}) = \{v_1, \ldots, v_{i-1}, v_{i+1}, \ldots, v_n\}$ and $v_k \in vars(\textit{eff}(\widehat{m}_k))$ (where $\widehat{m}_k \in creases(m, k)$) is an element of $D_{n-i}^{m:n}$ for each $i \in [0, n]$. Therefore, $D^{m:n}$ is a partition of $V^{\leq n}$.

Looking at a sub-sequence of the base macros of $m$, focusing on the base macros $[b_{g \cdot 2^{d+1}+1}, \ldots, b_{g \cdot 2^{d+1}+2^d}, \ldots, b_{g \cdot 2^{d+1}-1}] =: m_{d,g}$ for an arbitrary $d \in [0, n]$ and an arbitrary $g \in [0, 2^{n-d} - 1]$.

Since $m$ is folded $n$ times, the macro $m_{d,g}$ is folded $d$ times with $b_{g \cdot 2^{d+1}+2^d}$ as $d$-th crease.

With the Matching Lemma we conclude $s_{g \cdot 2^{d+1}+2^d-1-j}[\![b_{g \cdot 2^{d+1}+2^d}!]\!] = s_{(g+1) \cdot 2^{d+1}+2^d+j}$ for all $j \in [0, 2^d - 1]$. With index shift we get $s_{g \cdot 2^{d+1}+i}[\![b_{g \cdot 2^{d+1}+2^d}!]\!] = s_{(g+1) \cdot 2^{d+1}-1-i}$ for all $i \in [0, 2^d - 1]$. Since $d$ and $g$ were arbitrarily chosen it holds for all $g \in [0, 2^{n-d-1}]$ and for all $d \in [0, n]$.

With the Orthogonality Lemma we see, that $D_d^{m:n}$ does not contain any meta-variable created by a variable from $vars(\textit{eff}(b_{g \cdot 2^{d+1}+2^d}))$ and projects the difference away. Therefore

$$\forall d \in [0, n] \forall g \in [0, 2^{n-d} - 1] \forall i \in [0, 2^d - 1]:$$
$$s_{g \cdot 2^{d+1}+i}^{D_d^{m:n}} = s_{(g+1) \cdot 2^{d+1}-1-i}^{D_d^{m:n}}$$

We finally conclude that we can apply the $2^{n+1}$ states criterion to reveal a correlation complexity of at least $n + 1$. $\qquad\square$

The folded macro criterion with $n = 1$ is equivalent to the criterion from Theorem 1.

## Arbitrary Correlation Complexity

The $2^n$ states criterion and the folded macro criterion show sufficient conditions to detect a lower bound of the dimension of a potential heuristic. They do not answer the question if a planning task with arbitrary correlation complexity exists. In the following, we show that the answer is yes by applying the macro folding criterion on a Gray counter task with an arbitrary number $n$ of bits.

## Gray Counter Task

The Gray Counter Task with 3 bits is the example Seipp et al. (2016) provided with correlation complexity 3. It iterates through all binary numbers of a given length in an order such that consecutive numbers differ on only one bit, this is known as the Gray code (Gray 1953).

**Definition 6** (Gray Counter Task). *The gray counter task of $n$ bits is a planning task $\Pi_n = \langle V_n, I_n, O_n, \gamma_n \rangle$ with the state variables $V_n = \{v_0, \ldots, v_{n-1}\}$ with domain $\{0, 1\}$, the initial state $I_n = \{v_i \mapsto 0 | i \in [0, n-1]\}$, the operators*

$$
\begin{aligned}
O_n = \{ & \langle \{v_j \mapsto 0, v_{j-1} \mapsto 1\} \\
& \cup \{v_i \mapsto 0 \mid 0 \leq i < j-1\}, \{v_j \mapsto 1\}\rangle, \\
& \langle \{v_j \mapsto 1, v_{j-1} \mapsto 1\} \\
& \cup \{v_i \mapsto 0 \mid 0 \leq i < j-1\}, \{v_j \mapsto 0\}\rangle \\
& \mid 0 < j < n \} \\
\cup \{ & \langle \{v_0 \mapsto 0\}, \{v_0 \mapsto 1\}\rangle, \langle \{v_0 \mapsto 1\}, \{v_0 \mapsto 0\}\rangle \}
\end{aligned}
$$

*and the goal $\gamma_n = \{v_{n-1} \mapsto 1\} \cup \{v_i \mapsto 0 \mid i \in [0, n-2]\}$.*
*The task $\langle V, \gamma, O, I \rangle$ is a reverse gray counter task if $\langle V, I, O, \gamma \rangle$ is a gray counter task.*

We can also view it as a recursive construction. The gray counter task of 1 bit is a planning task $\Pi = \langle V, I, O, \gamma \rangle$ with the state variables $V = \{v_0\}$ with domain $\{0, 1\}$, the initial state $I = \{v_0 \mapsto 0\}$, the operators $O = \{\langle \{v_0 \mapsto 0\}, \{v_0 \mapsto 1\}\rangle, \langle \{v_0 \mapsto 1\}, \{v_0 \mapsto 0\}\rangle\}$ and the goal $\gamma = \{v_0 \mapsto 1\}$.
The gray counter task of $n$ bits is a planning task $\Pi$ with the state variables $V = \{v_0, \ldots, v_{n-1}\}$ (each with domain $\{0, 1\}$) that consists of two tasks $\Pi_{1/2}$, a gray counter task of size $n-1$ on the variables $\{v_0, \ldots, v_{n-2}\}$ with an additional constant state variable $v_{n-1} \mapsto 0$ and $\Pi_{2/2}$, a reverse gray counter task of size $n-1$ on the variables $\{v_0, \ldots, v_{n-2}\}$ with an additional constant state variable $v_{n-1} \mapsto 1$. The initial state of $\Pi$ is the initial state of $\Pi_{1/2}$, the single goal state of $\Pi$ is the single goal state of $\Pi_{2/2}$ and the operators of $\Pi$ is the set containing:

- each operator of $\Pi_{1/2}$ and $\Pi_{2/2}$,
- the operator with the goal state from $\Pi_{1/2}$ as precondition and the initial state from $\Pi_{2/2}$ as effect and
- the operator with the initial state from $\Pi_{2/2}$ as precondition and the goal state from $\Pi_{1/2}$ as effect.

By looking at the Gray counter task through the lens of recursion it is clear what our macros have to be. We choose $\overrightarrow{m}$ to be the macro that solves $\Pi_{1/2}$ and $\overleftarrow{m}$ the macro that solves $\Pi_{2/2}$, both have $2^{n-1} - 1$ base macros, namely the original operators. They are obviously $2^{n-1}-1$-matching as they solve inverse problems. By the recursive construction of the gray counter task $\overrightarrow{m}, \overleftarrow{m}$ are both folded $n-2$ times. This fits the condition of the folded macro criterion and reveals a correlation complexity of $n$ for the gray counter task on $n$ bits.

This shows that planning tasks of arbitrary correlation complexity do exist. Seipp et al. (2016) left an open question whether or not examples of "naturally occurring" planning domains that are tractable and contain tasks with high correlation complexity exist.

## Turing Machine

If one ignores the tractable part the question can be answered with yes. Each domain that can directly encode a Turing Machine (TM) of arbitrary memory size has unbounded correlation complexity.

What do we mean by encoding a finite tape Turing machine directly?

**Definition 7** (direct TM encoding). *A TM is defined as $\langle Z, z_0, z^*, \Gamma, \delta \rangle$ with $Z$ the set of internal states, $z_0, z^* \in Z$ where $z_0$ is the initial state and $z^*$ is the accepting state, $\Gamma$ the set of tape symbols, and $\delta : (Z \setminus z^*) \times \Gamma \to Z \times \Gamma \times \{-1, +1\}$ the transition function.*
*A planning task $\Pi$ encodes a TM directly if it is solvable and*

- *for each configuration $c$ that TM traverses there is a corresponding landmark state $s_c$,*
- *if TM traverses $c_1$ before $c_2$ then each plan traverses $s_{c_1}$ before $s_{c_2}$, and*
- *there is a subset $V^{tape}$ of $V$ such that:*
  - *$s_{c_1}^{V \setminus V^{tape}} = s_{c_2}^{V \setminus V^{tape}}$ for each configuration $c_1, c_2$ that TM traverses and the internal state and the head position of $c_1$ and $c_2$ are the same.*
  - *Each variable in $v_{t_i} \in V^{tape}$ corresponds to a cell $t_i$ of the tape and for each $c$ traversed by TM with symbol $a$ in cell $t_i$ the state $s_c$ contains the fact $v_{t_i} \mapsto d_{t_i,a}$.*

Only TMs that halt can be encoded directly. The entry of a single cell could be encoded by multiple state variables in $V^{tape}$ but each $v \in V^{tape}$ encodes only one cell. Since $V$ is finite the tape has to be finite, too. The initial state of the planning task encodes the initial configuration of the TM, including the input on the tape.

Bylander (1994) described (in Theorem 3.1) a way to transform a TM into a planning task. This planning task encodes said TM directly.

Consider a Turing machine that reads the input string, changes it to the next string according to the gray code, and sets the head back to the initial position. The machine repeats this until the final value of the gray code is represented by the tape. With an initial input of $n$-many 0's the correlation complexity of the task simulating such a Turing machine is at least $n$. This can be shown with the $2^n$ states criterion with the states that represent the Turing machine at the beginning of the described loop.

Many domains of the International Planning Competition (IPC) enforce each task to have a correlation complexity of at most 2 (Seipp et al. 2016). Culberson (1997) showed how to encode TMs into Sokoban tasks. This corresponds to a direct encoding, too.

Helmert (2006) showed an encoding of TMs into Promela tasks. However, this encoding does not fit our definition. There the content of the cell at the position of the head and the position of the head is encoded in one variable and is therefore not a *direct* encoding. The partition of $V$ is not possible. This is not the relevant hurdle, as we can still apply the Macro Folding Criterion on a Promela task that represents the described TM in Helmert's encoding. The two macros are putting the Promela-message that represents the

TM symbol $1/0$ into the Promela-queue that represents the cell which represents the second most significant bit on/off. The relevant hurdle is the PDDL encoding (Edelkamp 2003) as it allows to activate multiple Promela-transitions without executing them directly. This provides too much freedom to easily detect critical, matching macros.

In the 1st Combinatorial Reconfiguration Challenge (CoRe Challenge 2022) Christen et al. (2023) encoded the independent set reconfiguration (ISR) problem as a planning task. The graph track asked for an instance with a long solution. They provided one that encodes a Gray counter. We can apply the macro folding criterion on this instance, too. This reveals a correlation complexity of at least $n/5$ for their graph track submissions, where $n$ is the number of nodes in the instance graph.

The earlier described Termes task has correlation complexity of at least 3. We discussed that the primary set-up and the primary tear-down are critical macros. They are 3-matching and folded once. With that, we can apply the macro folding criterion to confirm a correlation complexity of at least 3.

This shows that we have occurrences of such domains (how natural these occurrences are is up to debate). Promela occurred in IPC4, Sokoban occurred in IPC6, ISR in the 1st CoRe Challenge, and Termes in IPC9.

The original question about tractable domains remains open but with the Folded Macro criterion, we have a tool to show that a task has high correlation complexity.

## Discussion

We were not able to find further IPC domains that allow tasks with a correlation complexity larger than 2. The way we found the ones we did was by constructing a task with only one (cycle free) solution. Without that, there is often some freedom (similar to the Promela domain) which hinders us from finding matching pairs of critical, folded macro.

Lacking further examples of IPC domains where we can apply our new criterions might be a downside for their relevance, on the one hand. On the other hand, it indicates that potential heuristics are often sufficiently expressive even with low dimensions. The criterion gave further insight into what is challenging for potential heuristics but also provides a possible approach to tackle such challenges, namely with macros. Investigating planning based on macros (Jonsson 2009) seems to be a good candidate to accompany low dimensional potential heuristics. Planning based on macros is strong in the shortcomings of low dimensional potential heuristics.

We found a generalization of the criterion from Theorem 1 for arbitrary correlation complexity. There might be a generalization to detect arbitrary correlation complexity for the criterion from Theorem 2, which incorporates dangerous operators. Such a criterion could provide further insight.

## Related Work

The most significant difference between the $\Pi^{\leq k}$ construction and the $P^m$ construction by Haslum (2009), the $\Pi^{\mathcal{C}}$ compilation by Steinmetz and Hoffmann (2018), or fluent merging by van den Briel, Kambhampati, and Vossen (2007) is that the $\Pi^{\mathcal{C}}$ construction does not describe a valid planning task. The operators are not fitting to the resulting state space (it would be possible to extend the definition in a way that we construct matching operators with metavariables in the precondition and effects. However, this would be rather cumbersome and unnecessary since we do not need the operators for our arguments).

The additional variables in a $\Pi^{\mathcal{C}}$ compilation have a binary domain and each additional variable represents a partial assignment. In other words, a conjunction of facts. The metafacts in the $\Pi^{\leq k}$ construction represent a combination of facts. The domain of such a metavariable is the cartesian product of the corresponding domains. The set of partial assignments that are considered by a $\Pi^{\mathcal{C}}$ compilation, is not further specified. If $\mathcal{C}$ contains all partial assignments of size $\leq k$, then we can interpret the additional facts from the $\Pi^{\mathcal{C}}$ compilation as the translation into STRIPS of the additional facts from $\Pi^{\leq k}$.

Fluent merging is defined on a finite domain representation and merging two variables combines their domains by a cartesian product like the $\Pi^{\leq k}$ construction. However, fluent merging replaces the variables it merges with the new one. Fluent merging reduces the number of state variables while the $\Pi^{\leq k}$ construction increases the number of state variables.

The $P^m$ construction is defined on propositional STRIPS tasks, while the $\Pi^{\leq k}$ construction and the $\Pi^{\mathcal{C}}$ compilation are defined on tasks in finite domain representation. However, the $P^m$ construction considers all partial assignments of size $\leq m$ and is in this regard similar to the $\Pi^{\leq k}$ construction.

## Conclusion

We have shown that the correlation complexity of a planning task can be arbitrarily large. This means no fixed dimension for potential heuristics, in combination with simple hill climbing, will be sufficient for satisficing planning on arbitrary domains. It is in some cases possible to detect a large correlation complexity with the newly introduced $2^n$ states criterion and Folded Macro criterion. We also showed that if a domain can encode a Turing machine in a certain way, then we can create tasks of arbitrary correlation complexity in this domain.

With the new criterions, we gained a deeper understanding of what structure causes a large correlation complexity.

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

# Appendix A

**Corollary 1.** *The quadruple criterion is a generalization of the criterion from Theorem 2.*

We show that the condition of the criterion from Theorem 2 implies the condition of the quadruple criterion.

*Proof.* The correlation complexity requires by definition a descending, dead-end avoiding heuristic.

If $\Pi = \langle V, I, O, \gamma \rangle$ is a planning task in normal form with the operator $o$ that is dangerous and critical in $\Pi$ then for each DDA heuristic $h$ there exist reachable states $s$, $s[\![o]\!]$, $s'$, $s'[\![o]\!]$ with $s$, $s[\![o]\!]$, $s'$ alive and $s'[\![o]\!]$ unsolvable and $o$ applicable in $s$ and $s'$ and $h(s) > h(s[\![o]\!])$ and $h(s'[\![o]\!]) \geq h(s')$.

Let $M := vars(pre(o))$ and $W := V \setminus M$. With $\Pi$ is in normal form we conclude $s^M = s'^M$ and $s[\![o]\!]^M = s'[\![o]\!]^M$. Because of $\mathit{eff}(o) \subseteq pre(o)$, applying the operator $o$ does not affect any variable in $W$. Therefore, $s^W = s[\![o]\!]^W$ and $s'^W = s'[\![o]\!]^W$.

It remains to show that $\{M, W\}$ is a partition of $V$. Therefore, we assume $vars(pre(o)) = V$. This implies that $pre(o)$ is the only state where $o$ is applicable and therefore $s[\![o]\!] = s'[\![o]\!]$ but the one is solvable while the other is unsolvable. We conclude that the assumption is wrong and that $vars(pre(o)) \subsetneq V$. Considering that $o$ is critical. This implies that $\emptyset \neq \mathit{eff}(o)$. Since $o$ is in normal form we know that $vars(\mathit{eff}(o)) \subseteq vars(pre(o))$ and therefore $\emptyset \subsetneq vars(pre(o))$. So with $M = vars(pre(o))$ we conclude $\emptyset \subsetneq M \subsetneq V$. Therefore, $\{M, W\}$ is a partition of $V$.

This shows that we can use the quadruple criterion, because for each heuristic $h$ that is DDA there exists states $s, s[\![o]\!], s'[\![o]\!], s'$ in $\Pi$ and a partition $\{W, M\} = V$ such that:

$h(s) > h(s[\![o]\!])$ and $h(s'[\![o]\!]) \geq h(s')$ and $s^W = s[\![o]\!]^W$ and $s'[\![o]\!]^W = s'^W$ and $s^M = s'^M$ and $s[\![o]\!]^M = s'[\![o]\!]^M$ $\qquad\square$