# OpenReview forum: "Higher-Dimensional Potential Heuristics: Lower Bound Criterion and Connection to Correlation Complexity"
_icaps-conference.org/ICAPS/2024/Conference — ICAPS 2024_

### Official Review · Reviewer_gM19 · 2024-01-19

**Significance And Importance:** 2
**Soundness:** 3
**Novelty:** 3
**Clarity:** 3
**Overall Evaluation:** 1
**Confidence:** 3

**Weaknesses:**

1: Minor weaknesses that are easily fixable.

**Contributions Of The Paper:**

----- post-rebuttal -----
Thank you for the detailed response!
-----------------------------


The paper provides new insights into the connection of potential heuristics and correlation complexity.
The main contribution of the paper is that it strictly generalizes existing criteria that can detect a correlation complexity of planning tasks that is 2 or greater.
The second contribution is the introduction of actual example tasks and domains with high/unbounded correlation complexity.

**Ethical Considerations:**

(1) Not Applicable: The paper does not have any ethical considerations to address

**Nomination For Best Paper:**

No

**Questions For Authors:**

1) Line 207-210: I don't understand that sentence. Can you elaborate please.
2) Proof of theorem 7:
Why does the fact that \overleftarrow{m} and \overrightarrow{m} are both critical imply that they must appear in every plan?
From the definition of critical, there only needs to exist two alive states s,s' which are unsolvable without one of the two macros.
I don't see why s=s'=I.

**Reproducibility:**

0: N/A - nothing to reproduce.

**Strengths Of The Paper:**

The paper contributes a novel understanding of the correlation complexity of classical planning tasks.
The analysis is novel and non-trivial, and presented in an accessible way.
The authors connect their analysis to example domains from the IPC, an encoding of Turing machines, and a recently used planning encoding of gray counters.

**Weaknesses Of The Paper:**

The write-up is very dense and sometimes hard to follow. Several parts could benefit from a bit of polishing.
While I believe that all claims are correct, I could not follow all details of some of the later proofs.
In particular the indexing of the macro-folding part is quite challenging.
I think the paper could highly benefit from a few examples and intuitions.

In my opinion, the current related work section does not add much value. While the \Pi^<k construction is central to the paper, it is just
a supporting concept and not a main contribution of the paper. As such, I don't see why you compare it to the \Pi^C or \P^M constructions.
What I would find a lot more interesting (but optional) is in how far the results of this paper (in particular existence of tasks with high correlation complexity) are implied, or at least indicated, by some of the results of Pommerening et al AAAI'17, and Steinmetz&Hoffmann IJCAI'18.
Is there any connection to context-dependency graphs? Assuming there was a way to learn a set of minimal-size conjunctions that yields h*,
does the maximal size of any conjunction imply the correlation complexity?


Minor stuff:
- criterions --> criteria?
- line 76: missing \setminus
- line 93: marcos
- line 116: capital P in subscript should probably be \cal
- line 135: what is S?
- top left of page 4: subscript of v_{robothand}

---

> ### Author Rebuttal · Authors · 2024-01-27
>
> R3:
> About L207-210
>
> A:
> Comparing two states with h_1 will provide the same order as comparing them with h_2.
> h_1(a)>h_1(b) iff h_2(a)>h_2(b) (same for >=).
> We will add this iff relation to the paragraph to prevent further misunderstanding.
>
> R3:
> Why does the fact that \l{m} and \r{m} are both critical imply that they must appear in every plan?
> From the definition of critical, there only needs to exist two alive states s,s' with are unsolvable without one of the two macros.
> I don't see why s=s'=I.
>
> A:
> Thank you for pointing that out, this is indeed an oversight. Both macros should be critical for the same alive state.
> The Thm is worded incorrectly and therefore stronger than what we meant. We will fix that for the camera-ready with
> "Let \Pi be a planning
> task in normal form, $a$ an alive state in \Pi, and let \r{m} and \l{m} be critical macros for $a$ that are 2
> (n − 1)-matching and folded n − 1 times, then \Pi
> has correlation complexity of at least n + 1."
> In the proof we replace $I$ with $a$, and "plan for \Pi" with "$a$-plan".
>
> We will also remove the comment about the equivalence to Thm 1 as it is not true with the weaker statement.
> The weaker statement does not interfere with the rest of the paper. All further arguments are based on macros that are critical for the initial state.
>
> @R1:
> About discussion
>
> A:
> You are right. It is not a necessary condition that they are applicable for a high correlation complexity.
> We will reformulate to "this strengthens our belief that dim=2 is often sufficient."
> In "An Empirical Study of Perfect Potential Heuristics" (ICAPS 2019)
> Augusto B. Corrêa, Florian Pommerening
> look at the size of features for nested potential heuristics that represent the perfect heuristic which is also DDA but even more restrictive.
> We are not aware of other work that provides the correlation complexity of a domain besides Seipp et al. 2016.
>
> @R1:
> About Termes partition
>
> A:
> The hard thing about this is realizing that the statement "The corresponding metastate I^{V^\leq2}
> contains 31 metafacts" is incorrect. Thank you for triggering this. The number is actually 16. We will fix that for the camera ready.
>
> Consider this renaming
> 2:v_{height2}
> 3:v_{height3}
> 4:v_{height4}
> a:v_{robotAt}
> h:v_{robotHand}
> and
> 23 means <v_{height2}, v_{height3}>.
>
> One partition for the Termes example is:
> D_2= { <.>, 2, a, h, 2a, 2h, ah   }
> D_1= {  3, 4, 23, 24, 34, 3a, 4a  }
> D_0= {  3h, 4h  }
> Other partitions are also possible.

---

### Official Review · Reviewer_cuBr · 2024-01-22

**Significance And Importance:** 2
**Soundness:** 4
**Novelty:** 3
**Clarity:** 3
**Overall Evaluation:** 2
**Confidence:** 4

**Weaknesses:**

1: Minor weaknesses that are easily fixable.

**Contributions Of The Paper:**

The paper focuses on computing correlation complexity of tasks, i.e., the
minimum dimension of potential heuristics that are descending and dead-end
avoiding (i.e., those that allow to reach a goal by the hill-climbing algorithm
simply following the decreasing h-values). Authors introduce new criteria for
finding correlation complexity which generalizes over the previously known
criteria. Moreover, they show there exist tasks with arbitrarily high
correlation complexity.

**Ethical Considerations:**

(1) Not Applicable: The paper does not have any ethical considerations to address

**Nomination For Best Paper:**

No

**Questions For Authors:**

1. Can Lemma 1 be strengthened as I mentioned before?

2. Can you comment on the "Turing Machine" section and explain what did you mean
by it?

3. Feel free to comment on anything else I mentioned in my review.

==== POST-REBUTTAL ===
Thank you for your responses.
Please, include in CR the changes you proposed in your responses to my and other reviews.

I would also suggest that you don't speak about "naturally occurring" domains and to be more clear/straightforward in the "Turing Machine" section. First, I don't think there is anything like "natural" domain. I certainly cannot imagine how would "unnatural" domain look like. Second, in the "Turing Machine" section you can simply argue that there are domains with unbounded correlation complexity like Sokoban, and then show that any domain that can encode the described TM will have unbounded correlation complexity. I think this is a decent contribution in its own right and there is no reason to confuse the reasoning by referring to some kind of natural domains.

**Reproducibility:**

0: N/A - nothing to reproduce.

**Strengths Of The Paper:**

The paper is relevant to ICAPS. Its focus is mainly theoretical -- it is a
continuation of previous works addressing the question how to measure complexity
of planning tasks. I am not aware of any practical applicability of this
approach, but I do not see it as an issue and I think this might come later,
especially with the newly proposed approach using macros.

I think all claims are correct. Although there are some minor (easily fixable)
issues with the formalism. I think the paper is a solid contribution. In
particular, the connection to macros is, in my opinion, non-trivial and not
obvious and it could lead to more interesting results later (especially in
combination with compilations like P^m, P^C and alike).

**Weaknesses Of The Paper:**

There are some issues that should be addressed for camera-ready. I will list
them mostly in order as they appear in the paper (severity of the issues is, I
think, self-evident):

Abstract should not use citations.

The notation X^V is never introduced. I could infer from the context that it
means restriction of a (partial) state X to variables V, but it should be
explicitly stated.

Definition of pre([o_1,o_2]) is missing set operator between pre(o_2) and
eff(o_1) -- I assume it should be pre(o_2) \setminus eff(o_1).
More importantly, macro is not well-defined, because it completely leaves out
the question of applicability of the sequence of operators it consists of (and
the application of operator/macro on a state is defined only for applicable
operators/macros). This should be explicitly defined. Either allow macros only
over sequences of operators that are applicable in the intermediate states, or
allow application of operators that are not applicable (it sounds weird, but one
could say that o applied on s is s[o] = s if pre(o) \not\subseteq s).

Page two, third paragraph: "The set pre([o_1,o_2]) might be no partial state."
I do not understand what it means. Maybe it has something to do with the issue
above? I guess if o_2 is not applicable after o_1, then
(pre(o_2) \setminus eff(o_1)) \cup pre(o_1) might contain two facts from the
same variable -- is this the case? If so, I suggest resolving it by simply
allowing only "meaningful" macros.

There is a mix-up between the definition of heuristic mapping *states* to
numbers, and potential heuristics mapping *partial states* to numbers. The
simplest solution I see is to define heuristics also over partial states and
then clearly explain why it is needed (after all, this would also make the
following text easier to understand).

Explain what Iverson bracket means -- I guess saying it is a notation for a
characteristic function would be enough.

Def. 1 defines correlation complexity for domains, but domains are never
formally introduced. In fact, a domain in this context is very informal concept
because correlation complexity reported for a domain (for example in
previous papers on correlation complexity) is not necessarily correlation
complexity over all possible tasks with the same PDDL domain. The reason is that
PDDL domains often allow to construct tasks that are not what the modeller had
in mind when writing the PDDL (that are not in the "spirit" of the domain). For
example, one could construct logisitics tasks where a truck is at multiple
locations at the same time as so on. In any case, this part should be removed
from the formal definition and moved to the text below as a sidenote (it's
actually not even important for the paper).

The paper could be improved with more examples, starting with examples for
Theorem 1 and 2. There seems to be enough space for it as some parts are
unnecessarily long (more on that below).

Page 2, last paragraph before the proof in right column. I do not see why waste
space for explaining to readers that sum over X is equal to sum of sums over its
partitions. It is so obvious that I would not include it at all -- I believe
every reader with a high-school diploma is able to figure this out.

The conditions in Theorem 3 and Theorem 4 are the same -- I suggest to give it a
speaking name or introduce a notation for it and then reuse it. It will not only
shorten the text, but it will also make it easier for the reader to match these
two theorems.

Proof of Theorem 3: The last step can be easily shortened to something like
Since a^M = d^M and b^M = c^m it follows that h^pot(a^M) > h^poy(b^M) \geq h^pot(a^M),
which is a contradiction.
There is also stray dot after h^pot(a^M).

Paragraph below the proof of Theorem 3: There is no need to pose rhetorical
questions, just state what you want to state: The quadruple criterion is
sufficient but not necessary condition...

Proof of Theorem 4 is unnecessarily long, because it is essentially just a
simple corollary of Theorem 3.

The title "2^n states criterion" should be capitalized.
There is no need for the first sentence of "2^n states criterion" subsection.

Def. 3:
 - Why is are subscripts of metavariables tuples instead of sets? The order of the
indexes does not matter (it matters for values though), and it would make more
clear that important is the combination of variables (not all orderings).

 - The definition meta^{\leq k}(s) should be
meta^{\leq k}(s) = \{meta_S(s) \mid S\subseteq V, |S|\leq k\}
not the \bigcup notation because meta_S(s) is a single pair (metafact).

 - "metafacts from s" does not make sense because there are no metafacts in s
   (s consists of facts).

 - shouldn't metafact be written as v \mapsto d instead of tuple <v,d>?

Last paragraph on page 3: It mentions constants -- they are not introduced nor
used in any way in the formalism. What do you mean by "without the constants"?

Isn't Lemma 1 unnecessarily weak? Unless I'm missing something, *for every*
potential heuristic of dimension k there exists a k-synchronized one of
dimension 1.

Theorem 5: \mathbb{N}_1 is not defined, I assume it suppose to mean
n\in\mathbb{N} and n >= 1. If so, please, write it as such.

All over the paper: I find it hard to read things like "n - 1-synchronized". I
think it would be easier to read if you used parenthesis: (n - 1)-synchronized.

Again, condition in Theorem 5 and Theorem 6 is the same -- please, give it a
name or introduce a notation for it (especially since it is very non-trivial
condition).
The proof of Theorem 6 is again unnecessarily long -- it is a simple corollary
of Theorem 5.

Def. 5: It is hard to distinguish arrow <- and -> above m because it's so small.
Could you use some notation that is easier to read?

Proof of Theorem 7: Please, replace "Matching Lemma" and "Orthogonality Lemma"
with Lemma 3 and 2, respectively. It would much easier to see what is referred.

The description of Gray Counter task says that the planning task \Pi consists of
two planning tasks \Pi_1/2 and \Pi_2/2. This is inaccurate (tasks here don't
consist of tasks), I guess what was meant by it is that \Pi can be decomposed
into two tasks \Pi_1/2 and \Pi_2/2.

The first sentence of "Turing Machine" section refers to the last sentence of
the previous section -- please, rephrase is.

I don't understand what is the point of the whole "Turing Machine" section.
It certainly does not answer the question about "naturally occurring" domains
with high correlation complexity. First, it is not clear what "naturally occurring"
actually means, so we can hardly try to answer this question on a formal
(technical) level. Second, TMs can encode any task, naturally occurring or not,
so I'm at loss what supposed to be point of this section. It definitely does not
make very convincing argument that there exist some kind of naturally occurring
tasks with high correlation complexity.

---

> ### Author Rebuttal · Authors · 2024-01-27
>
> R2:
> About pre([o_1,o_2])
>
> A:
> see rebuttal to Reviewer1
>
> R2:
> About domains in Def.1
>
> A:
> Yes, this is indeed not necessary. We will remove this from the formal definition as you suggested.
> We see a domain simply as a set of tasks. The set of tasks that fit a given domain.pddl file and the set of tasks that fit the spirit of that domain.pddl file might be different and their CC might be different, too.
>
>
> R2:
> The conditions in Theorem 3 and Theorem 4 are the same -- I suggest to give it a
> speaking name or introduce a notation for it and then reuse it.
>
> A:
> Thank you for this suggestion. We will implement that into the camera-ready version.
>
> "If there exists [a witnessing quadruple] then the dimension of $h^{pot}$ is at least 2."
>
> "If there exists [a witnessing $2^n$-tuple] then the dimension of $h^{pot}$ is at least n."
>
>
> R2:
> Def. 3:
>  - Why is are subscripts of metavariables tuples instead of sets?
>
> A:
> There is indeed no need for an order in the index of the metavariable. We will change this as you suggested.
>
>
> R2:
> About Lemma 1
>
> A:
> You are not missing something. This is as we meant it. We will reformulate it to avoid confusion.
>
>
> R2:
> Can you comment on the "Turing Machine" section and explain what did you mean
> by it?
>
> A:
> Let us assume a reader sees Sokoban as a naturally occurring domain.
> We want to show that it has an unbounded correlation complexity. The TM is a stepping stone for that. The argument is that we can create a Sokoban task that encodes any size-bounded TM (Culberson 1997) and we choose a TM that encodes an n-bit gray counter. With that setup of nested encoding, we can apply the 2^n states criterion on such a Sokoban task.

---

### Official Review · Reviewer_uzUT · 2024-01-22

**Significance And Importance:** 3
**Soundness:** 3
**Novelty:** 3
**Clarity:** 3
**Overall Evaluation:** 2
**Confidence:** 4

**Weaknesses:**

1: Minor weaknesses that are easily fixable.

**Contributions Of The Paper:**

Correlation complexity is the size of the conjunctions that need to be considered by a potential heuristic in order to solve a planning task without doing search. This paper provides new criteria to determine lower bounds of correlation complexity in order to prove that a given planning task has a correlation complexity of at least n, and shows that there exist planning tasks with correlation complexity larger than 2.

**Ethical Considerations:**

(1) Not Applicable: The paper does not have any ethical considerations to address

**Nomination For Best Paper:**

Yes

**Questions For Authors:**

What is the partition in the Termes example? Is it hard to lay out?

**Reproducibility:**

0: N/A - nothing to reproduce.

**Strengths Of The Paper:**

The paper is a very nice piece of research, containing multiple interesting theoretical results that advance our understanding of potential heuristcs for satisficing planning. The results are novel, significant and relevant for the ICAPS community. As far as I could check, the paper is technically correct and all the results are well proven. I found really interesting the connection between macros and correlation complexity/potential heuristics.

The paper is generally well-written (except the introduction, as I point out below). The weaknesses are very minor compared to the strengths so the paper should be accepted.

**Weaknesses Of The Paper:**

The paper organization is not ideal, as the contributions and structure of the paper is not communicated to the reader, so each new Section comes a bit as a surprise. For example, when reading the background, the reader is introduced to macros. But at that point is very unclear why macros would be relevant here.

An important part of this problem is the introduction, which does state some facts, but does not convey important information:
 * Motivation: The introduction says what correlation complexity is, but it does not say anything about why this is an interesting concept. The entire paper goes about how to show that there exist planning tasks with high correlation complexity. But, why should we care? Why is this finding interesting?

 * Contributions: The introduction does not say anything about what are the contributions of the paper. It would be important to mention that:
    - Seipp et al. already introduced a way to prove that tasks have a correlation complexity of at least 2.
    - This paper introduces new criteria. First, two criteria based on properties of the states. The quadruple criterion generalize the previous criteria being able to show correlation complexity of at least 2 in more cases. Then, the 2n criterion further generalizes the quadruple criterion to prove a correlation complexity of at least n.
      And then, new criteria based on the operators of the tasks are provided to avoid reasoning about a large amount of states. The new criteria are based on macros.
    - Then, we show a family of planning tasks and, using our criteria we show that correlation complexity grows with the task size, so there exist tasks with arbitrary correlation complexity.

 I'd have appreciated having some summary along those lines in the introduction so I'd recommend including that in the final version.



Technical correctness:

 * In Lemma 2, it is not mentioned that the task is in normal form, but that is assumed within the proof.

 * The proofs are slightly hard to follow and, while I understood the main arguments, I found hard to follow some of the details:

   - line 512: the proof uses the notation D^n:m. I don't understand the meaning of n:m, and given that n and m are used with many different meanings over the paper, perhaps they could be ommited here and just use D or other symbol?

   - In Theorem 7, it'd be nice to explain the partition that is used in the Termes example. As later it is mentioned that such partition exists but it is never provided.

   - In Definition 6, when j=0, the precondition v_{j-1} is not defined.

   - In the section "Turing Machine" several IPC domains are discussed. "Turing machine" is not a very good name for such a subsection. The claim saying that ISR instances have n/5 correlation complexity does not have  too many supporting arguments. Is this analyzed in more detail in some appendix?

 * In the discussion section, the paper says that the fact that the current criteria are not applicable indicates that low dimensions are sufficient in most domains. I don't think this is the case, as perhaps correlation complexity is high even if the criteria cannot prove it. Perhaps it'd be good to remind the reader the known results (for how many domains we know their correlation complexity is 1, 2, or > 2? for how many domains is the question still open?)

Minor comments:

- The plural of criterion is criteria.

- Line 23: 'measures' -> 'measure'.
- Line 76 '$pre(o_2)eff(o_1)$': a missing operator in between it seems. Also, why exactly is pre([o1,o2]) not a partial state?
- Line 90: 'might be no partial state' -> add 'called empty macro' or something like that,
  since 'non-empty macros' is used in the following sentence without defining it.
- Theorems 1 and 2 -> missing dot at the end.
- Line 259: `if $t \in l$`: $t$ and $l$ not defined.
- Table 1, line 282, line 285: Use `\mathit`, `\mbox` or something else in subindexes and 'free/full' for a better look.
- Line 285 (beginning of the page): '$v_robotHand$' -> '$v_{robotHand}$'.
- Line 286: 'contains 31 metafacts' -> it would be useful to specify how is this number computed. Also, some subset of the 31 facts is given, but it is not clarified that this is just an example. If there is a reason to include those as example and not others, it should be explicitly said.
- Line 287: '$v_{\langle\rangle} \mapsto \langle \rangle$' -> this notation has not been introduced.
- Definition 6: Readability would be increased by specifying preconditions and effects of operators separately.

---

> ### Author Rebuttal · Authors · 2024-01-27
>
> About pre([o_1,o_2]), also @Reviewer2
>
> We would like to resolve this issue by fixing the pre([o_1,o_2]) definition such that pre([o_1,o_2]) \subseteq s iff o_1 is applicable in s and o_2 is applicable in s[[o_1]].
>
> With that, we can treat macros like operators. In particular, macro m is applicable in s if pre(m) \subseteq s.
>
> We will do that by defining
> pre([o_1,o_2]):=
> 	pre(o_1) \cup (pre(o_2) \setminus eff(o_1)) \cup { v \mapsto d \in eff(o_1) \mid v \in vars( pre(o_1) \cap pre(o_2) )}
>
> The 3rd set of this union prevents [o_1,o_2] from being applicable if o_1 "consumes" the common precondition of o_1 and o_2.
>
> Reasoning:
> if o_2 is not applicable in s[o_1] then there is a fact v_2 -> d_2 in pre(o_2) but v_2 -> d_x in s[o_1].
> The fact v_2 -> d_x is either in s or in eff(o_1) but not in both due to normal form. In the former case v_2 -> d_2 will be in pre([o_1,o_2]) because of (pre(o_2) \setminus eff(o_1)). In the latter case v_2 -> d_x will be in pre([o_1,o_2]) because of { v \mapsto d \in eff(o_1) \mid v \in vars( pre(o_1) \cap pre(o_2) )}.
> In both cases pre([o_1,o_2]) results in a set that is no partial state. Therefore by definition not applicable.
>
> In other words
> - pre(o_1) \subseteq pre([o_1,o_2]) ensures that: [o_1,o_2] applicable in s implies o_1 applicable in s.
> - pre(o_2) \setminus eff(o_1) \subseteq pre([o_1,o_2]) ensures that: [o_1,o_2] applicable in s implies o_2 applicable in s[o_1].
> - (pre(o_2) \setminus eff(o_1)) \cup { v \mapsto d \in eff(o_1) \mid v \in vars( pre(o_1) \cap pre(o_2) )} \subseteq pre([o_1,o_2]) ensures that: [o_1,o_2] *not* applicable in s implies o_2 not applicable in s[o_1].
>
> If the operators are in normal form the resulting macro is in normal form, too.
>
> This is similar to the definition of macros in
> "Online Identification of Useful Macro-Actions for Planning,
> Andrew Coles, Maria Fox and Amanda Smith ICAPS, 2007" (we will add this to the references),
> with the difference that they work with a STRIPS like representation and we work with SAS+ and that they allow combining operators to a macro with precondition and add/del-effects if they are applicable in succession, where we ensure that
> our resulting macro is not applicable if the operators are not applicable in succession.
>
> [For space reasons other answers to Reviewer1 are in the rebuttal to Reviewer3]

---

### Meta-Review · Area_Chair_NFZS · 2024-02-01

**Recommendation:** Accept (Oral)
**Confidence:** 5

**Metareview:**

This paper introduce a new method for obtaining a lower bound on the correlation complexity of planning tasks - which is the minimum dimension of a potential heuristic necessary to solve the task without search. The paper provides strong theoretic contributions and analyses examples using the new lower bound.

The write-up of the paper should be improved for the final version. Especially the introduction and the "Turing Machine" section w.r.t. the discussion of what is a natural vs unnatural task should be rephrase to be more clear. Further, I would recommend that the authors include the examples and information given in their rebuttal into the paper.

I (and the reviewers) appreciate the author's detailed rebuttal.

**Ethical Considerations:**

(1) Not Applicable: The paper does not have any ethical considerations to address